# QMCTorch: Molecular Wave Function with Neural Components for Energy and Force Calculations

**Nicolas Renaud[1*]**

1 Netherlands eScience Center, Matrix THREE Science park 402, 1098 XH Amsterdam, The Netherlands

* nicolas.gm.renaud@gmail.com

## Abstract

In this paper, we present results obtained using QMCTorch, a modular framework for real-space Quantum Monte Carlo (QMC) simulations of small molecular systems. Built on the popular deep learning library PyTorch, QMCTorch is GPU-native and enables the integration of machine learning-inspired components into the wave function ansatz, such as neural network backflow transformations and Jastrow factors, while leveraging efficient optimization algorithms. QMCTorch interfaces with two widely used quantum chemistry packages - PySCF and ADF - which provide initial values for atomic orbital exponents and molecular orbital coefficients. In this study, we present wavefunction optimizations for four molecules: $H_2$, $LiH$, $Li_2$, and $CO$, using various wavefunction ansätze. We also compute their dissociation energy curves and the corresponding interatomic forces along these curves. Our results show good agreement with baseline calculations and recover a significant portion of the correlation energy. QMCTorch provides a modular and extendable platform for rapidly prototyping new wavefunction ansätze, evaluating their performance, and analyzing optimization outcomes.

## Contents

31

32

## 1  Introduction

34  The electronic structure of molecular systems is typically determined by solving the time-
35  independent Schrödinger equation: $\mathcal{H}\Psi(\mathbf{r}) = E\Psi(\mathbf{r})$ where $\Psi(\mathbf{r})$ represents the many-body
36  wave function, $\mathbf{r}$ denotes the electronic coordinates and $\mathcal{H}$ is the molecular Hamiltonian:

$$\mathcal{H} = -\frac{1}{2}\sum_i \nabla_i^2 + \sum_{i>j}\frac{1}{|\mathbf{r}_i - \mathbf{r}_j|} - \sum_i\sum_\alpha\frac{Z_\alpha}{|\mathbf{r}_i - \mathbf{R}_\alpha|} + \sum_{\alpha>\beta}\frac{Z_\alpha Z_\beta}{|\mathbf{R}_\alpha - \mathbf{R}_\beta|} \tag{1}$$

37  where indexes $i, j$ run over the electrons and $\alpha, \beta$ over the nuclei. The first term in eq. (1),
38  $\nabla_i^2 = \frac{d^2}{dx_i^2} + \frac{d^2}{dy_i^2} + \frac{d^2}{dz_i^2}$, represents the kinetic energy of the $i$-th electron. The second term
39  accounts for electron-electron repulsion while the third describes electron-nucleus attraction,
40  with $Z_\alpha$ the atomic number of the $\alpha$ atom. The last term corresponds to the repulsive inter-
41  action between nuclei. According to the variational principle, the ground-state solution of the
42  Schrödinger equation can be approximated by minimizing the variational energy:

$$E_v = \frac{\int \Psi^*(\mathbf{r})\mathcal{H}\Psi(\mathbf{r})d\mathbf{r}}{\int |\Psi(\mathbf{r})|^2 d\mathbf{r}} \geqslant E_0 \tag{2}$$

43  that reaches the ground state energy $E_0$ when $\Psi$ exactly matches the ground state wave func-
44  tion of the system. Solving equation (2) becomes increasingly challenging when the number of
45  electrons in the system increases. Quantum Monte-Carlo (QMC) simulations offer an power-
46  ful alternative to mean-field and post Hartree-Fock (HF) approaches, providing high-accuracy

approximations of the variational energy. In QMC simulations, the high-dimensional integral in eq. (2) is evaluated through Monte Carlo integration [1]:

$$E_v = \int \rho(\mathbf{r}) E_L(\mathbf{r}) d\mathbf{r} \approx \frac{1}{M} \sum_{k=1}^{M} E_L(\mathbf{r}_k) \tag{3}$$

where $\mathbf{r}_k$ are electronic positions, sampled from the probability distribution $\rho(\mathbf{r}) = |\Psi(\mathbf{r})|^2 / \int |\Psi(\mathbf{r})|^2 d\mathbf{r}$. The evaluation of the local energies at the sampling points: $E_L(\mathbf{r}) = \mathcal{H}\Psi(\mathbf{r})/\Psi(\mathbf{r})$ are independent from each other, enabling massive parallel execution. This characteristic makes QMC methods particularly well-suited for exascale computing environments [2].

Several high-performance software packages have been developed to solve the Schrödinger equation of molecular systems using a QMC approach [3–5]. These tools have been successfully applied to a broad range of computational chemistry problems, including excited state calculations [6–8], geometry optimization [9] and molecular dynamics simulations [10]. For a comprehensive overview of QMC methodologies and their applications, we refer the reader to Ref. [11]. In recent years, significant progress has been made in integrating machine learning techniques into QMC simulations [12–20]. In these approaches, the wave function is represented by an artificial neural network, with variational parameters optimized via automatic differentiation. By leveraging modern deep learning frameworks such as PyTorch [21], TensorFlow [22], or JAX [23], these methods can efficiently utilize hardware accelerators, significantly enhancing computational performance.

Two notable machine learning-inspired approaches, Ferminet [24] and Paulinet [25], have demonstrated remarkable accuracy in computing ground-state energies of molecular systems. Both methods are capable of recovering 99% or more of the correlation energy across a diverse set of molecules. Ferminet employs a wave function ansatz that enforces only the antisymmetry requirement of the wavefunction, delegating the task of learning atomic and molecular orbital structure entirely to the neural network. In contrast, Paulinet incorporates molecular orbitals obtained from conventional quantum chemistry calculations and augments them with neural network components to capture electronic correlation effects more effectively. These approaches have since been extended to excited-state calculations [26–28], underscoring the growing impact of machine learning techniques in advancing quantum chemistry.

We introduce QMCTorch [29], a real-space Quantum Monte Carlo (QMC) framework for small molecular systems, built on the PyTorch deep learning library [21]. In QMCTorch, the wave function is represented using a Slater-Jastrow ansatz, where atomic orbitals, molecular orbitals, Jastrow factors, and optionally backflow transformations are implemented as differentiable layers. The parameters of these components are optimized to minimize the variational energy using gradient-based methods. QMCTorch interfaces with PySCF [30] and ADF [31] to initialize atomic orbital exponents and molecular orbital coefficients. Efficient kinetic energy estimators, originally developed by Filippi et al. [32,33], are extended here to support backflow-transformed wave functions, replacing standard automatic differentiation techniques. Additionally, QMCTorch enables the evaluation of interatomic forces by treating atomic coordinates as variational parameters within the wave function ansatz. This capability opens the door to applications such as geometry optimization and molecular dynamics simulations.

## 2 Methods

### 2.1 Wave Function Ansatz

Like many QMC codes, QMCTorch employs a Slater-Jastrow wave function ansatz defined as:

$$\Psi_\theta(\mathbf{r}) = e^{\mathcal{J}(\mathbf{r})} \cdot \sum_n d_n \, D_n^\uparrow(\tilde{\mathbf{r}}^\uparrow) \, D_n^\downarrow(\tilde{\mathbf{r}}^\downarrow) \tag{4}$$

that satisfies the anti-symmetry principle $\Psi(r_0, ..., r_i, ...r_j, ...r_N) = -\Psi(r_0, ..., r_j, ...r_i, ...r_N)$. In eq. (4), $\mathcal{J}(\mathbf{r})$ denotes the Jastrow factor and $D_n^{\uparrow\downarrow}(\mathbf{r}^{\uparrow\downarrow})$ are the Slater determinants corresponding to spin-up and spin-down electrons respectively. The symbol $\theta$ represents the set of variational parameters included in the wavefunction. Fig. 1 illustrates the structure of the wave function ansatz. Starting from the electronic coordinates, a backflow transformation is used to capture electronic correlations [11]. This transformation is defined as: $\tilde{\mathbf{r}}_i = \mathbf{r}_i + \sum_{j \neq i} K_{\mathrm{BF}}(r_{ij})(\mathbf{r}_i - \mathbf{r}_j)$, where $K_{\mathrm{BF}}$ is the backflow kernel and $r_{ij}$ denotes the inter-electronic distance. A variety of backflow kernels have been proposed over the years, ranging from many-body formulations [34], to orbital-dependent kernels [35] and more recently to flexible neural kernels [36–39]. QMCTorch includes several built-in backflow kernels, such as inverse-distance kernels and fully connected feedforward neural network kernels. Additionally, users can define custom backflow transformations by implementing their own function for $K_{\mathrm{BF}}(r_{ij})$. All necessary derivatives for computing the local energies can be automatically obtained via automatic differentiation, allowing users to rapidly prototype and test new backflow kernels without the need to manually derive and implement complex analytical expressions.

The backflow-transformed electronic positions are used to evaluate the atomic orbitals (AOs) of the molecular system. QMCTorch includes a fully differentiable atomic-orbital layer that computes the AO values at the given electronic coordinates. This layer supports both Gaussian and Slater Type Orbitals (STOs) whose initial exponents - and, when applicable, contraction coefficients - are provided by PySCF [30] or ADF [31], respectively. In the following, we exclusively use STOs as they naturally respect electron-nucleus cusp conditions. STOs are given by: $\varphi_\alpha(\tilde{r}_i) = \mathcal{N} x^{k_x} y^{k_y} z^{k_z} \tilde{r}^{k_n} e^{-\zeta_\alpha |\tilde{r}_i|}$, where $x, y, z$ are the Cartesian components of the vector between the electron at $\tilde{r}_i$ and the center of the basis function, $k_m$ are integers determined by the quantum numbers of the atomic orbitals and $\mathcal{N}$ is a normalization constant. The exponents $\zeta_\alpha$ control the spatial extent of the orbitals and are treated as a variational parameters within the wavefunction ansatz. As illustrated in Fig. 1, the AO layer also takes the atomic coordinates as input. This design enables the computation of the local energy derivatives with respect to atomic positions via automatic differentiation, an essential feature for evaluating interatomic forces.

The molecular orbitals (MOs) are constructed as a linear combination of AOs following the expression: $\phi_\alpha(\tilde{r}_i) = \sum_\beta (c_{\beta,\alpha} \times \mu_{\beta,\alpha}) \varphi_\beta(\tilde{r}_i)$, where $c_{\beta,\alpha}$ are the molecular orbital coefficients obtained from a Hartree–Fock calculation and remain fixed during optimization. The $\mu_{\beta,\alpha}$ are variational scaling parameters introduced in the MO layer, all initialized to 1. This formulation enables the optimization of molecular orbitals while preserving their original symmetries, in contrast to directly optimizing the Hartree–Fock coefficients $c_{\beta,\alpha}$, which can mix orbitals of different symmetries.

Slater determinants are computed from MO matrices corresponding to spin-up and spin-down electrons, based on the electronic configurations specified by the user. Once these determinants are calculated, a fully connected layer perform the weighted sum defined in eq. (4).

The coefficients $d_n$ in this expansion are treated as variational parameters, all initialized to 0 except for the ground-state determinant, which is assigned an initial weight of 1. In contrast to other approaches [25], QMCTorch does not require a preliminary calculation to determine the most relevant determinants to include in the expansion.

The Jastrow factor is computed using the original electronic coordinates based on the following expression [40]:

$$J(\boldsymbol{r}) = \sum_{i<j} K_{\text{ee}}(r_{ij}) + \sum_{\alpha,i} K_{\text{en}}(r_{\alpha i}) + \sum_{\alpha,i>j} K_{\text{een}}(r_{\alpha i}, r_{\alpha j}, r_{ij}) \tag{5}$$

where $K_{\text{ee}}$, $K_{\text{en}}$ and $K_{\text{een}}$ are the electron-electron, electron-nucleus and electron-electron-nucleus Jastrow kernels. Not all the terms in the Jastrow factor need to be included; for instance, the wavefunction ansatz can be restricted to contain only an electron–electron kernel. QMCTorch provides several built-in kernels, and users can easily define and test their own custom kernels. As with the backflow transformation, all necessary derivatives for computing the local energies can be obtained via automatic differentiation. This enables rapid experimentation with new forms of the Jastrow factor without requiring manual derivation of lengthy analytical expressions. The wave function, as defined in Eq. (4), is obtained as the product of the Jastrow factor and the sum over the Slater determinants.

## 2.2 Gradients of the Wave Function

The variational parameters of the wave function ansatz are optimized by minimizing the loss function: $\mathcal{L}(\theta) = {}^1\!/\!_M \sum_m E_L(r_m)$. The gradients of the loss function with respect to the variational parameters are computed using a low-variance estimator originally described in Ref. [41].:

$$\nabla_\theta \mathcal{L} = \frac{1}{M} \sum_m 2 \frac{\nabla_\theta \Psi_\theta(\mathbf{r}_m)}{\Psi_\theta(\mathbf{r}_m)} (E_L(\mathbf{r}_m) - E_v) \tag{6}$$

This expression only requires computing the derivatives of the wave function with respect to the variational parameters ($\nabla_\theta \Psi_\theta(\mathbf{r}_m)$) which can be efficiently obtained via automatic differentiation. This estimator avoids the need to differentiate the local energy itself. The evaluation of the local energies requires computing the diagonal elements of the Hessian of the wave function with respect to the electronic coordinates ($\nabla_i^2 \Psi(r)$). We employ an efficient numerical scheme which, in the absence of a backflow transformation, expresses the diagonal elements of the Hessian for both ground and excited state determinants as traces of matrix products [32]. We have extended this approach to accommodate backflow-transformed wave functions, significantly accelerating the computation of local energies (see Appendices F, G, and H).

## 2.3 Interatomic Forces

Interatomic forces can be calculated using the wave function ansatz depicted in Fig. 1 as the atomic positions are input parameters of the model. The interatomic forces are evaluated using the estimator [10]:

$$\mathcal{F}_\alpha = -\frac{1}{M} \sum_{m=1}^{M} \nabla_\alpha E_L(\mathbf{r}_m) + (E_L(\mathbf{r}_m) - E_v) \nabla_\alpha \ln \rho(\mathbf{r}_m) \tag{7}$$

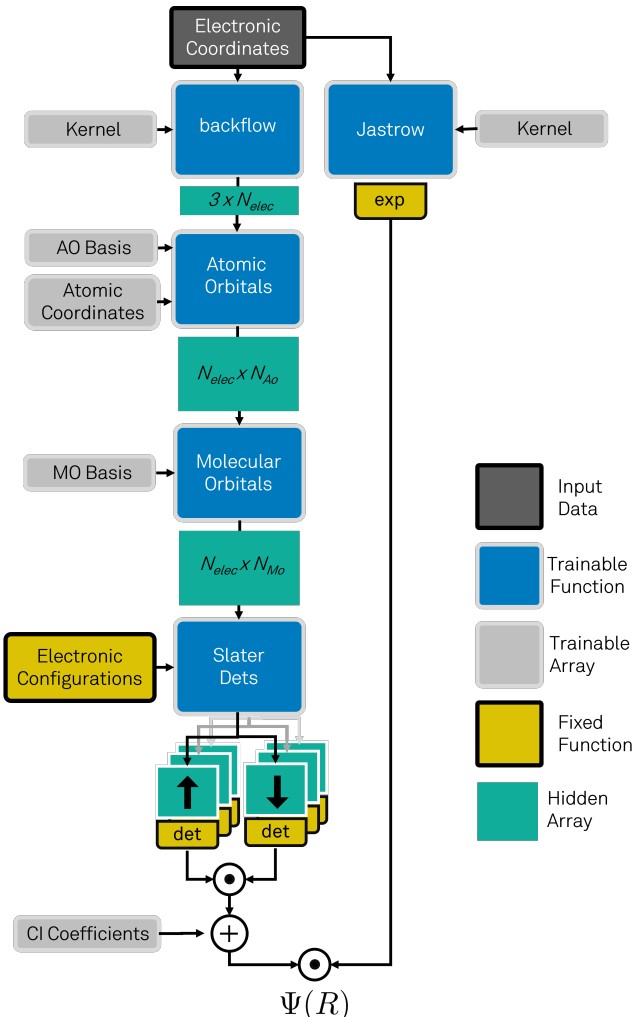

Figure 1: Representation of the wave function ansatz in QMCTorch. The architecture consists of multiple layers that transform electronic and atomic coordinates into a scalar wave function value. All layers are fully differentiable, enabling efficient optimization of the variational parameters via gradient-based methods. Additionally, certain layers support user-defined kernels, allowing for fine-tuning and customization of the ansatz.

that only requires the gradients calculation of the local energies ($\nabla_\alpha E_L$) and logarithmic density ($\nabla_\alpha \ln \rho (\mathbf{r}_m)$) with respect to the atomic coordinates. It is important to note, however, that this estimator exhibits infinite variance when applied to approximate wave functions, necessitating a large number of samples for reliable convergence. To address this limitation, regularization techniques have been proposed to improve the estimator's stability and efficiency, and their implementation is planned for future work [42, 43].

## 2.4 Computational Performance

Leveraging PyTorch's capabilities, QMCTorch can compute the wave function, energy, loss function gradients, and interatomic forces on both CPUs and GPUs. As demonstrated in Appendix A, GPU acceleration becomes effective only when the number of sampling points is sufficiently large. For instance, with 1000 sampling points for a CO molecule, the GPU run-

183   time is one to two orders of magnitude lower than that on a CPU (see Fig. 5). The runtime
184   scales favorably with the number of determinants included in the wave function, owing to the
185   efficient numerical schemes for evaluating ground and excited-state determinants. Notably,
186   the computation of $\nabla_\theta \mathcal{L}$ and $\mathcal{F}_\alpha$ incurs only a marginal overhead compared to the evalua-
187   tion of local energies alone, especially when the wave function includes a large number of
188   determinants.

## 3   Results

190   We demonstrate the capabilities of QMCTorch on four small molecules: $H_2$, $LiH$, $Li_2$ and $CO$.
191   We use here a simple inverse backflow transformation given by the kernel: $K_{BF}(r_{ij}) = \omega/r_{ij}$
192   with $\omega$ the single variational parameter of this transformation. We use an electron-electron
193   neural Jastrow factor whose kernel, $K_{ee}(r_{ij})$, is given by two fully connected feedforward neu-
194   ral networks - one for parallel-spin and one for antiparallel-spin electron pairs. Each network
195   contains two hidden layers of 32 and 64 nodes respectively and uses a sigmoid as activation
196   function. Additional details regarding the optimization of the wave functions are provided in
197   Appendix B.

### 3.1   Wavefunction Optimization

199   Fig. 2 shows the correlation energy retrieved during the optimization of the four molecules
200   using different wave function ansätze. SD-NJ: a single determinant with a neural Jastrow but
201   no backflow; MD-NJ; multiple determinants with a neural Jastrow but no backflow; MD-NJ-
202   iBF: multiple determinants with a neural Jastrow and an inverse backflow. For each ansatz,
203   simulations were performed both with and without optimization of the atomic orbital expo-
204   nents. The retrieved correlation energy is defined by: $\Delta E = 1 - (E_v - E_{HF})/(E_{ref} - E_{HF})$ with
205   $E_{HF}$ ($E_{ref}$) the Hartree-Fock(Reference) energies reported in Table 1 and $E_v$ the variational en-
206   ergy obtained with QMCTorch.

208       As shown in 2, the results obtained with the three different ansätze give satisfying results
209   on all four molecules. In most cases, the total energy initially lies above the Hartree–Fock
210   energy but quickly converges toward the reference value during optimization. Depending on
211   the molecule and ansatz, the optimization recovers between 50% and 99% of the correlation
212   energy. Optimizing the atomic orbital exponents consistently improves the variational energy.
213   Notably, a comparison of the first two columns in Fig. 2 reveals that the variational energy
214   obtained with the SD-NJ ansatz and optimized AOs is comparable to that of the MD-NJ ansatz
215   with frozen AOs. This suggests that computational frameworks currently lacking AO optimiza-
216   tion could benefit from extending the variational space to include differentiable AOs, thereby
217   enhancing the accuracy of the resulting wave function.

219       The neural Jastrow factors used in SD-NJ and MD-NJ ansätze are, in principle, capable of
220   approximating any arbitrary functions. However, as illustrated in Fig. 6, their optimized form
221   closely resemble a Padé-Jastrow factor, a widely used functional form in QMC simulations. As
222   a result, replacing the neural Jastrow factor by a much simpler Padé-Jastrow factor yields qual-
223   itatively similar performance (see Fig. 7). It is interesting to note that the electron-electron
224   cusp conditions of the neural Jastrow factors, i.e. $\frac{dJ(x)}{dx}\big|_{x=0}$, fluctuate around their theoretical
225   limits during the optimization. While the neural Jastrow is more flexible than a Padé-Jastrow
226   factor, and could in principle lead to lower variational energy values, it suffers from its inability
227   to exactly respect these cusp conditions. Incorporating these conditions directly into the neu-

|        | $E_{\mathrm{HF}}$ (a.u.) | $E_{\mathrm{ref}}$ (a.u.) | $\Delta E_{\mathrm{QMCTorch}}$ | $\Delta E_{\mathrm{PauliNet}}$ | $\Delta E_{\mathrm{Ferminet}}$ |
|--------|--------------------------|---------------------------|--------------------------------|--------------------------------|--------------------------------|
| $H_2$  | -1.133                   | -1.17447 [51]             | 99.12%                         | 99.99%                         | -                              |
| $LiH$  | -7.97                    | -8.07055 [52]             | 93.57%                         | 99.30%                         | -                              |
| $Li_2$ | -14.867                  | -14.9954 [53]             | 89.21%                         | -                              | 99.47%                         |
| $CO$   | -112.7871                | -113.3225 [24]            | 63.47%                         | -                              | 99.32%                         |

Table 1: Ground state energy and fraction of the correlation energy recovered by QMCTorch, Paulinet [25] and Ferminet [24] for four different molecules.

ral architecture through hard constraints [44], may help overcome this limitation in the future.

The third column of Fig. 2 presents results obtained by augmenting the MD-NJ ansatz with an inverse backflow transformation. Consistent with previous findings [25], the inclusion of a backflow transformation introduces significant noise into the optimization process and generally fails to improve performance. A similar trend is observed in Fig. 7, where a more flexible neural backflow transformation is used. Interestingly, the neural backflow in Fig. 7 tends to converge toward a $1/x$ functional form during the optimization (see Fig. 8). This suggests that while neural backflows offer flexibility, simpler forms may be sufficient to obtain similar performance. A different approach consists of incorporating the backflow transformation through a differentiable envelope function, as implemented in Ref. [25] using the SchNet architecture [45]. This approach has demonstrated greater stability and is planned for future integration into QMCTorch.

The Jastrow factors employed in the preceding analysis include only electron-electron interaction terms. It is well established that incorporating many-body terms can significantly enhance the accuracy of variational energy estimates [40]. Figure 7 illustrates that, in our case, the addition of electron-nucleus and electron-electron-nucleus Jastrow kernels yields only a marginal improvement in the computed energy values. These results, along with those presented earlier, suggest that the limitations of our approach may not stem from the expressiveness of the ansatz itself, but rather from the optimization strategy or the sampling techniques used to evaluate the local energies and the gradients of the loss function.

As shown in Table 1, the maximum correlation energy retrieved with QMCTorch remains lower than that obtained with Paulinet and Ferminet. However it is important to remember that the ansätze used here involved significantly fewer variational parameters - on the order of $\sim 10^2$ to $10^3$ for QMCTorch - compared to $\sim 7 \cdot 10^4$ for Paulinet and $\sim 7 \cdot 10^5$ for Ferminet. Additionally, our optimization was explicitly limited to 500 steps, whereas Paulinet and Ferminet used up to $7 \times 10^3$ and $10^5$ steps, respectively. We also employed a simple ADAM [46] optimizer in contrast to the more advanced AdamW [47] and KFAC [48] optimizers used in Paulinet and Ferminet. Furthermore, our results may be constrained by the quality of the sampling procedure. We employed a standard Metropolis algorithm while more sophisticated variants are often adopted [49] and many studies refine the sampling points using diffusion Monte Carlo techniques [50].

## 3.2  Wavefunction and Energy Landscape of $H_2$

Fig. 3 shows the wavefunction and energy landscape of a $H_2$ molecule as its two electrons are independently moved along the molecular axis. As shown on this figure, the wavefunction naturally satisfies the electron-nucleus cusp conditions due to the use of STOs, resulting in

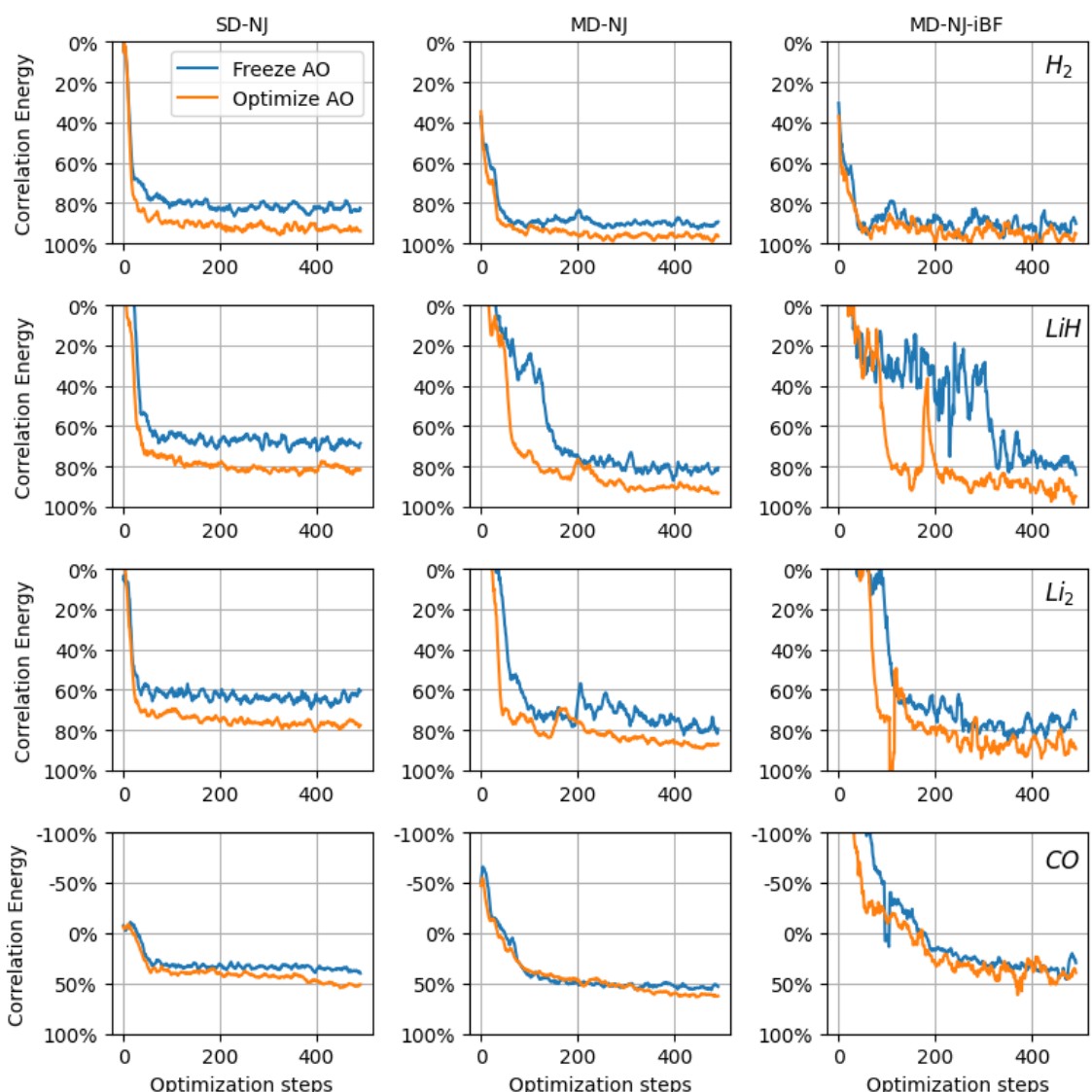

Figure 2: The y-axis indicates the fraction of the correlation energy recovered during the optimization process. The orange curves correspond to cases where all variational parameters were optimized, while the blue curves represent optimizations in which the exponents of the atomic orbitals were held fixed at their initial values. For clarity, all curves have been smoothed using a sliding window of size 10.

finite local energy values at the electron-nucleus coalescence points. One-dimensional profiles of the wavefunction and energy of $LiH$ and $Li_2$ are provided in Fig. 9. These plots confirmed that the wavefunction respects the electron-electron and electron-nucleus cusp conditions, as well as the Fermi exclusion principle. The wavefunction profile obtained for $LiH$ closely resembles that produced by Paulinet. However, the simpler nature of the ansatz used in our approach yields a smoother energy profile compared to Paulinet, albeit at the cost of reduced expressiveness.

## 3.3 Energy Curve and Interatomic Forces

To further assess the applicability of our approach, we have computed the dissociation energy curves of the four molecules studied here. The calculations were carried out using a multi-

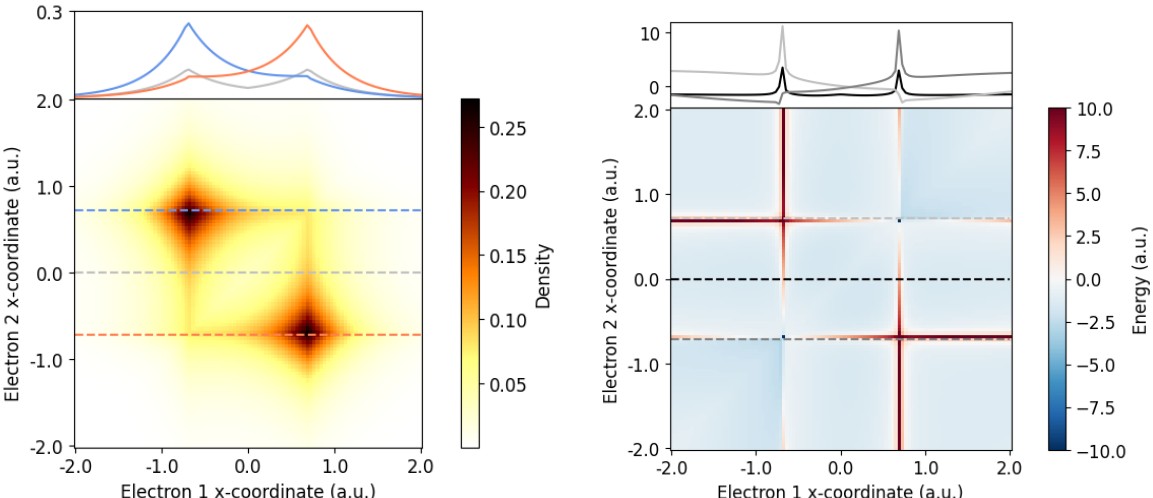

Figure 3: Values of $|\Psi(r_1, r_2)|^2$ (left) and $E(r_1, r_2)$ (right) of an optimized $H_2$ molecule as a function of the position of its two electrons along the axis of the molecule.

determinant wavefunction without any backflow transformation. Figure 4 presents the outcome of these simulations and compares them with simulations performed at the Hartree–Fock and coupled-cluster singles and doubles (CCSD) levels, using the PySCF package with a DZP basis set. As shown on this figure, the energy curves obtained with QMCTorch are in very good agreement with the CCSD results, even yielding slightly lower energies for $H_2$, $LiH$ and $Li_2$. However, as previously observed in Figure 2, QMCTorch recovers only about 60% of the correlation energy for CO. Consequently, in this case, our results do not outperform those obtained with CCSD.

Figure 4 also displays the interatomic forces computed along the dissociation curves, as evaluated using Eq. (7). To compute these forces, the optimized wavefunction was resampled using $10^6$ sampling points. These samples were divided in 100 batches and explicitly symmetrized around the molecular axis. The force component along the molecular axis (taken here as the $z$ axis), defined as: $\Delta z = F_z^\alpha - F_z^\beta$, with $F_z^\alpha$ the force along the z-axis for atom $\alpha$, was computed for each batch. The mean value and standard deviation of these values are shown in Fig. 4. As shown on this figure, the forces computed for $H_2$ and $LiH$ are in very good agreements with those obtained at the CCSD level of theory. However, the accuracy of the forces predicted by QMCTorch decreases for $Li_2$ an $CO$, particularly at larger interatomic separations, where the standard deviation increases significantly.

## 4  Conclusion

In this paper, we presented results obtained with QMCTorch, our software for performing real-space quantum Monte Carlo (QMC) simulations of molecular systems using PyTorch. QMCTorch is built around a modular implementation of a backflow-transformed Slater–Jastrow wavefunction. Unlike many similar codes, QMCTorch supports the use of Slater-type orbitals and enables the optimization of atomic orbital exponents through a fully differentiable atomic-orbital layer. The software includes efficient numerical schemes for computing determinants and kinetic energy, allowing it to handle large numbers of determinants with relatively low

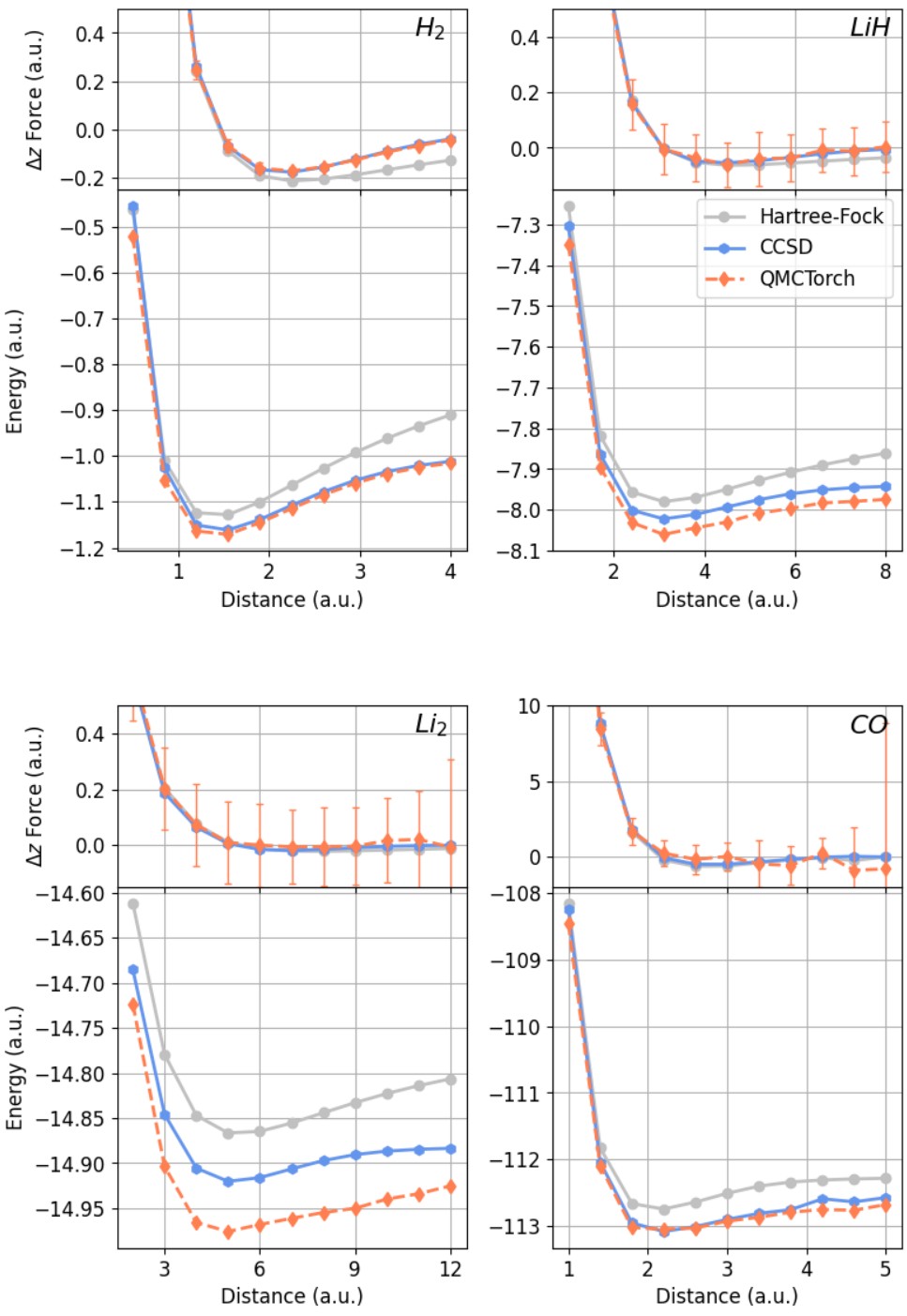

Figure 4: Dissociation energy curve and interatomic force ($\Delta z$) of four small molecules as a function of their interatomic distance computed with Hartree-Fock, CCSD and QMCTorch. Error bars on the interatomic forces represent the standard deviation across sampling batches.

303    computational cost.

304

305    The results obtained for four small molecules demonstrate that our approach is capable of
306    recovering a significant fraction of the correlation energy, despite the simplicity of the wave-
307    function ansatz. We have shown that optimizing the exponents of the atomic orbitals plays a

crucial role in improving wavefunction quality. Additionally, our findings indicate that while neural Jastrow factors enhance the flexibility of the ansatz, they do not significantly outperform the simpler, widely used Jastrow forms. Incorporating hard physical constraints into the architecture of neural Jastrow factors may further improve their effectiveness in future work. The inclusion of backflow transformations, as currently implemented in QMCTorch, led to noisy optimization and, consequently, a degradation in performance. Our results demonstrate that QMCTorch is able to produce accurate dissociation energy curves for all four molecules studied, underscoring the robustness of the approach. We also computed interatomic forces along these dissociation curves and found them to be in good agreement with CCSD reference values.

Currently, QMCTorch is limited to small molecular systems due to the absence of parallelization across multiple computing nodes. Extending the software to large-scale computing infrastructures through data parallelism is a key focus of future development. This enhancement will enable the simulation of larger systems, where QMC methods are often more efficient than alternative approaches [2]. Another major direction for future work is the integration of diffusion Monte Carlo (DMC) sampling techniques [54], which will allow for a more comprehensive assessment of the method's capabilities when combined with neural wavefunctions [55, 56]. We also plan to extend QMCTorch to support the optimization of excited-state wavefunctions [7, 8, 57], a feature already available in other neural wavefunction frameworks [26–28].

## Data Availability

QMCTorch is freely available under Apache 2.0 License at https://github.com/QMCTorch/QMCTorch. The input files used to run the simulations as well the output files and plotting scripts used for the different figures can be found at: https://zenodo.org/records/15583384

## Acknowledgments

We would like to express our gratitude to Prof. Claudia Filippi for guidance throughout the development of the software and to Felipe Zapata for collaboration in the early stage of the project.

**Funding information**  The development of the code was funded through the Joint NWO Computational Science and Energy Research & eScience Research Programme via the project *A Light in the Dark: quantum Monte Carlo meets solar energy conversion* grant ID: CSER.JCER.022.

## A   Computational Performance

Fig. 5 presents the GPU and CPU runtime for the calculation of the wave function ($\Psi$), local energies ($E_L$), gradients of the loss function ($\nabla_\theta \mathcal{L}$) and interatomic forces ($\mathcal{F}_\alpha$) for a *CO* molecule. The calculations were performed using varying number of determinants and either 10 or 1000 sampling points. As shown in this figure, no GPU acceleration is observed when using 10 points; in fact CPU calculation can be faster in this case. However, with 1000 sampling points, the GPU demonstrates a clear advantage, achieving a speedup of approximately one

348 order of magnitude for the wave function computation and nearly two orders of magnitude
349 for the energy, gradient, and force evaluations.

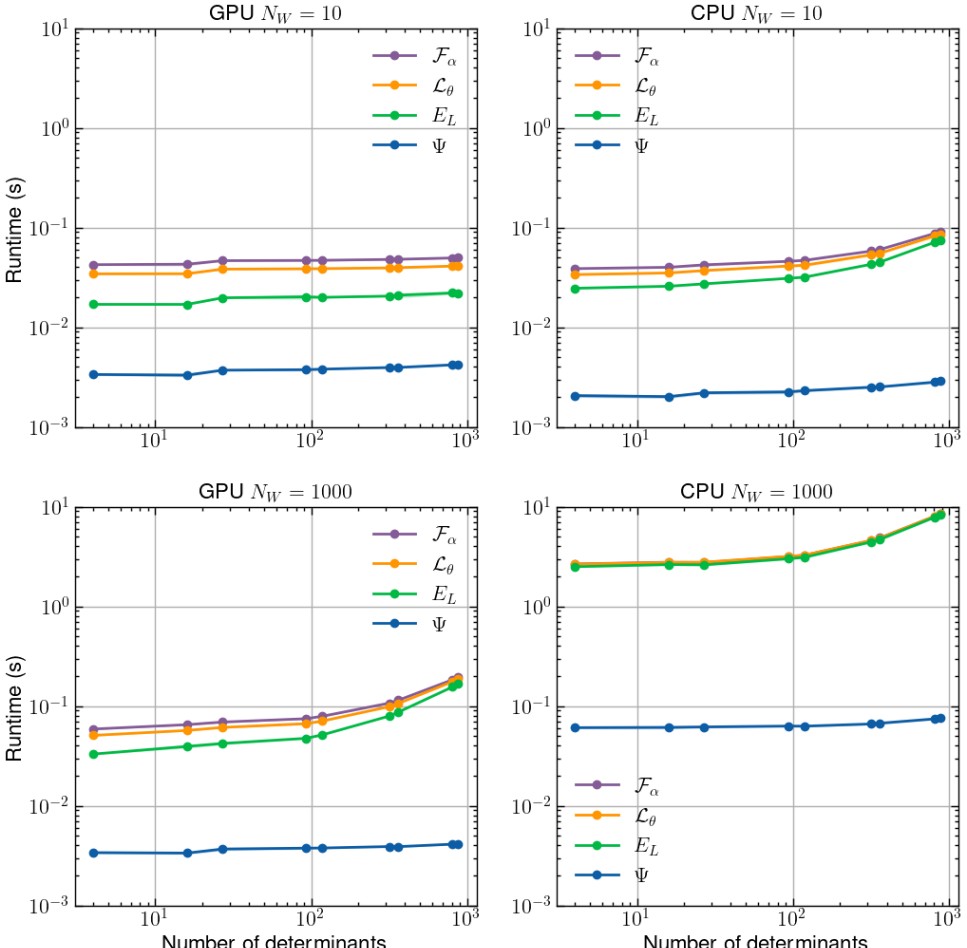

Figure 5: Runtime performance for computing the wave function ($\Psi$), local energies ($E_L$), gradients of the loss function ($\nabla_\theta \mathcal{L}$) and interatomic forces ($\mathcal{F}_\alpha$) of a CO molecule on an Nvidia A100 GPU and an AMD Rome 7H12 CPU. The calculations employed an electron-electron Padé-Jastrow factor and an inverse backflow transformation. Simulations were conducted across various electronic configurations, ranging from SD(2,2) to SD(10,10), corresponding to 4 to 876 determinants. Results are shown for two sampling regimes: 10 points (top) and 1000 points (bottom).

## B  Parameters for the Wavefunction Optimization

351 We provide here additional details regarding the wavefunction optimization procedure em-
352 ployed for the four molecules studied in this article. Table 2 summarizes all parameters used
353 during the optimization process.

354 **Wavefunction**  The initial HF calculations were done using ADF [31] with a DZP basis set.
355 Various electronic configurations were used in defining the wavefunction ansatz, ranging from
356 $SD(2,8)$ for $H_2$ to $SD(8,12)$ for $CO$. Depending on the molecule and the chosen ansatz, the
357 resulting wave functions comprised between 182 and 6906 variational parameters.

**Sampling** The sampling of $\rho(\mathbf{r})$ was done using a Metropolis-Hasting algorithm [58]. A total of $10^4$ independent walkers were employed, each performing $10^4$ Monte-Carlo (MC) steps. The walkers were moved using a normal proposal distribution with a standard deviation of 0.05 Bohr. All the electrons were moved simultaneously at each MC step. Only the final positions of the walkers were retained as sampling points for evaluating the local energies and gradients of the loss function.

**Optimization** The ADAM optimizer [46] was used to optimize the variational parameters using a learning rate of $10^{-2}$ and 500 optimization steps were performed. After each optimization step, all the sampling points were updated by resampling the probability distribution, using 500 MC steps.

| Parameter | Value(s) |
|---|---|
| SCF Package | ADF |
| Basis | DZP |
| Electronic configuration | |
| $H_2$ | SD(2,8) - (64 dets) |
| $LiH$ | SD(4,12) - (531 dets) |
| $Li_2$ | SD(4,12) - (531 dets) |
| $CO$ | SD(8,12) - (1425 dets) |
| Number walkers | $10^4$ |
| Number of initial MC steps | $10^4$ |
| Sampling step size | 0.05 |
| Resample Frequency | 1 |
| Number of resampling MC steps | 500 |
| Optimizer | ADAM |
| Learning rate | $10^{-2}$ |
| Number of optimization steps | 500 |
| Number of variational parameters | |
| $H_2$ | low: 182 - high: 4662 |
| $LiH$ | low: 857 - high: 5343 |
| $Li_2$ | low: 1189 - high: 5669 |
| $CO$ | low: 2425 - high: 6905 |

Table 2: Parameters used for the wavefunction optimizations shown in Fig. 2 and 7. The lower and upper bounds on the number of variational parameters correspond to the e-e Pade-Jastrow (Fig. 7) and MD-NJ (Fig. 2) ansatzes respectively.

# C  Neural Electron-Electron Jastrow Factor

The left panel in Fig. 6 shows the optimized form of the neural Jastrow factor obtained with a MD-NJ wavefunction ansatz depicted in Fig. 2. For clarity, these functions were normalized to their value at $x = 0$. While neural Jastrow factors are, in principle, capable of approximating arbitrary functions, they exhibit a form similar to that of the Padé–Jastrow factor (see Equation D.1).

The right panel in Fig. 6 illustrates the evolution of the electron-electron cusp of the

Jastrow factor, defined as: $\frac{dJ(x)}{dx}|_{x=0}$, throughout the optimization process. It is interesting to note that in most cases, the neural Jastrow factor converges toward the correct electron-electron cusp conditions given by [59]:

$$\frac{dJ(x)}{dx}\Bigg|_{x=0} = \frac{1}{4} \quad \text{for parallel spins and} \quad \frac{dJ(x)}{dx}\Bigg|_{x=0} = \frac{1}{2} \quad \text{for opposite spins} \quad \text{(C.1)}$$

However, these constraints are not explicitly enforced in the design of the neural Jastrow factor. As a result, there is no guarantee that they will be approached—let alone satisfied—by the optimized Jastrow function.

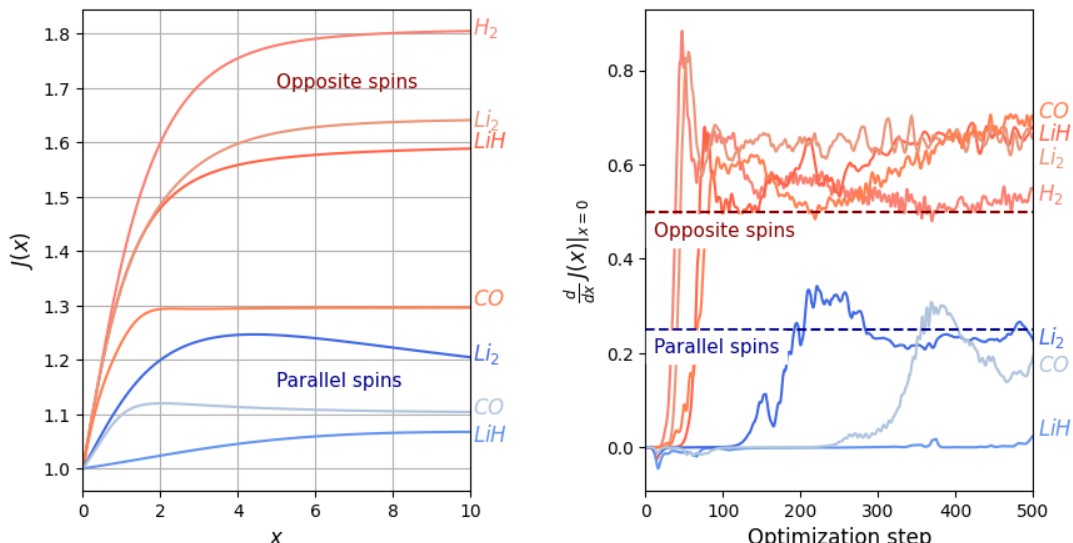

Figure 6: Left - Optimized form of the neural Jastrow factors. Right - Evolution of the electron-electron cusp conditions for the neural Jastrow factor throughout the optimization process. The dashed lines represent the theoretical values of 0.25(0.5) for parallel(opposite)-spins electron pairs.

# D    Additional Wavefunction Ansatz

Fig. 7 presents the correlation energy obtained with three different wavefunction ansatzes for the four molecules studied here.

## D.1    Electron-Electron Padé Jastrow Factor

In the first wave function considered here, we employed a simple electron-electron Padé-Jastrow factor defined as:

$$K_{ee}(r_{ij}) = \frac{\omega_0 r_{ij}}{1 + \omega r_{ij}} \quad \text{(D.1)}$$

where $\omega_0 = 0.25(0.5)$ for parallel(antiparallel)-spin electron pairs and $\omega$ a variational parameter. The fraction of the correlation energy recovered using this Jastrow factor is represented in the first column of Fig. 7. As shown in this figure, the results are not significantly different from those obtained with the neural Jastrow factor and presented in Fig. 2.

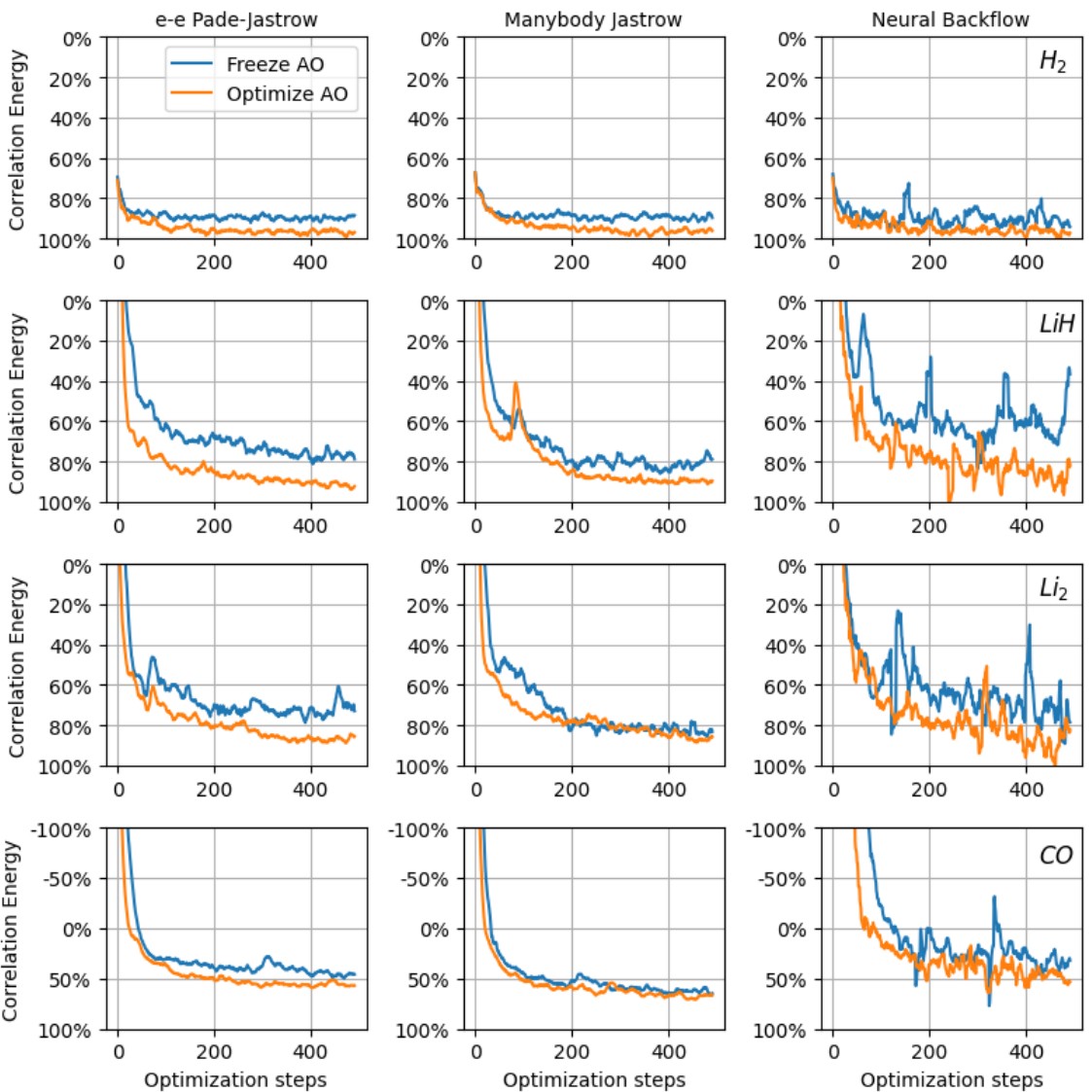

Figure 7: Optimization of the wave function of small molecules: $H_2$, $LiH$, $Li_2$ and $CO$ using different ansatzes (see text for details)

## D.2 Many-body Jastrow Factor

A many-body Jastrow factor was incorporated in the second wavefunction ansatz considered here. In addition to the electron-electron Padé-Jastrow kernel, we introduced an electron-nucleus kernel given by [60]:

$$K_{en}(r_{\alpha i}) = \frac{\omega_0 r_{\alpha i} + \omega_1 r_{\alpha i}^2}{1 + \omega_2 r_{\alpha i}} \tag{D.2}$$

with $\omega_0$, $\omega_1$ and $\omega_2$ variational parameters and $r_{\alpha i}$ the distance between atom $\alpha$ and electron $i$. We also introduce an electron-electron-nucleus kernel given by:

$$K_{een}(r_{\alpha i}, r_{\alpha j}, r_{ij}) = \sum_{\mu=1}^{N} c_\mu \left( \frac{a_{1_\mu} r_{\alpha i}}{1 + b_{1_\mu} r_{\alpha i}} \right) \left( \frac{a_{2_\mu} r_{\alpha j}}{1 + b_{2_\mu} r_{\alpha j}} \right) \left( \frac{a_{3_\mu} r_{ij}}{1 + b_{3_\mu} r_{ij}} \right) \tag{D.3}$$

where the coefficients $a_{n_\mu}$, $b_{n_\mu}$, $c_\mu$ are variational parameters. We used here $N = 5$ terms in the expansion. The fraction of the correlation energy obtained with this Jastrow factor

is represented in the second column of Fig. 7. As seen in this figure, the introduction of many-body Jastrow factor does not significantly changes the results obtained with an electron-electron Jastrow factor as shown in Fig. 2.

### D.3 Neural Backflow Transformation

The third wavefunction examined here, we used a simple Padé-Jastrow electron-electron factor but complemented the ansatz with a neural backflow transformation based on radial basis function network [61]. The kernel of this backflow transformation can be expressed as:

$$K_{BF}(r_{ij}) = \sum_{n=1}^{N_{RBF}} \omega_n \mathcal{N}_{\mu_n, \sigma_n}(r_{ij}) \tag{D.4}$$

with $\mathcal{N}$ the normal function of mean $\mu_n$ and std $\sigma_n$. $\omega_n$, $\mu_n$ and $\sigma_n$ are here a variational parameters. We used here $N_{RBF} = 10$ basis function in the expansion. The fraction of the correlation energy recovered using this Jastrow factor is shown in the third column of Figure 7. As illustrated, incorporating a neural backflow yields results that are qualitatively similar to those presented in Figure 2. However, the optimization process remains challenging, preventing the correlation energy from fully stabilizing.

The evolution of the backflow kernel $K_{BF}(r_{ij})$ during the optimization is shown in Fig. 8. As seen there, the values of the kernel function converges toward a form that is proportional to $1/r_{ij}$. This explain the limited impact that this neural backflow has on the final results compared to the ones obtained with an inverse backflow kernel.

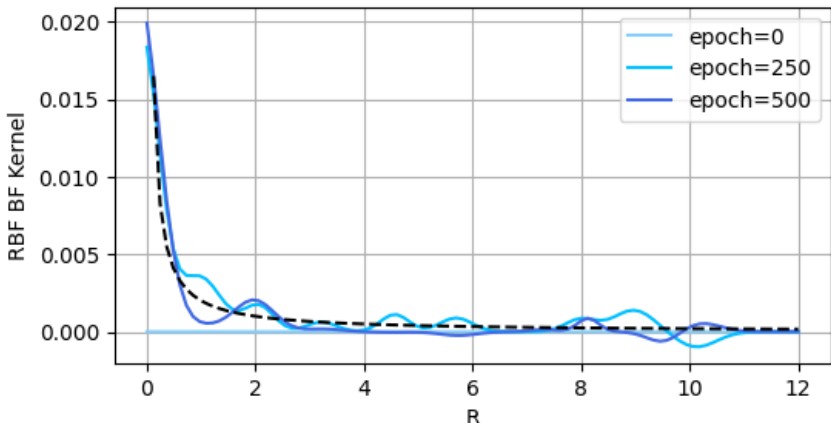

Figure 8: Evolution of the RBF neural backflow kernel during the optimization of the wavefunction.

## E Wavefunction and Energy Profile of $LiH$ and $Li_2$

Fig. 9 shows the wavefunction and energy profile of $LiH$ and $Li_2$, before and after the optimization of the wavefunction. The x-axis corresponds to the location of a spin-up electron that moves along the molecular axis (taken here as the z-axis). The positions of the nuclei and static electrons are marked on the figure. For $LiH$, the final spin-down electron is located at $(x, y, z) = (0.5, 0.5, 0.0)$, to enable direct comparison with the results reported in [25]. As shown on this figure, the wavefunction vanishes when two electrons with the same spin

coincide, in accordance with the Pauli exclusion principle. When two electrons of opposite spin occupy the same position, the wavefunction presents the correct electron-electron cusp behavior. Additionally, the electron-nucleus cusp conditions are well satisfied, due to the use of STOs in the calculation.

The wavefunction profile of $LiH$ closely resembles that obtained with Paulinet [25]. Interestingly, the wave function undergoes only minor changes during the optimization process, underscoring the subtle yet essential adjustments required to reach the variational minimum. The energy profile obtained with QMCTorch is noticeably smoother and more regular than that reported in Rf. [25]. This can be attributed to the simpler wavefunction ansatz employed in this work. However, as previously discussed, this simplicity also limits the ansatz's flexibility, which in turn constrains its ability to fully capture the true variational minimum

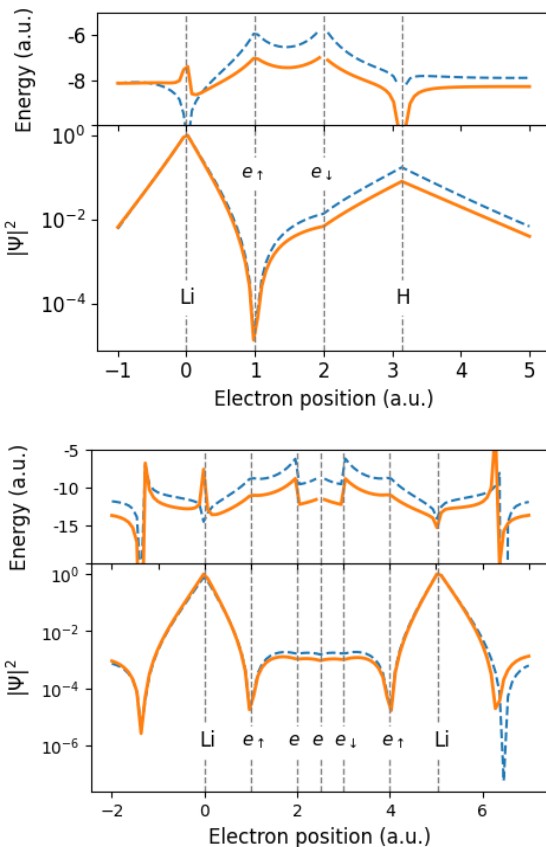

Figure 9: Wavefunction and energy profile of $LiH$ (top) and $Li_2$ (bottom) with respect to the position of a spin-up electron moved along the molecular axis. The dashed line represents the original wavefunction while the orange plain line represents the optimized wavefunction. The positions of the atomic centers and static electrons are marked in the figure for clarity.

# F    Computation of the Local Energies of Backflow-Transformed Wavefunctions

While all the derivatives of the wave function and the loss function could be obtained directly via automatic differentiation, we exploit here several numerical schemes to decrease the com-

441 putational and memory requirement of the calculations.

## F.1 Derivative of a Single Determinant

443 The local energy $E_L(\mathbf{r})$ requires the calculation of the kinetic energy $\frac{\nabla^2\Psi(\mathbf{r})}{\Psi(\mathbf{r})}$. Back-propagating
444 twice though the wave function via automatic differentiation can evaluate these terms but this
445 approach is computationally costly. To alleviate this issue we have implemented the results
446 obtained by Filippi et. al. [32] and extended them to the case of backflow transformed wave
447 function.
448

449 Setting $\Psi(\mathbf{r}) = \mathcal{J}(\mathbf{r})\Sigma(\mathbf{r})$, where $\Sigma(\mathbf{r}) = \sum_n d_n D_n^\uparrow D_n^\downarrow$, the kinetic energy can be written as:

$$\frac{\nabla^2\Psi(\mathbf{r})}{\Psi(\mathbf{r})} = \frac{\nabla^2\mathcal{J}(\mathbf{r})}{\mathcal{J}(\mathbf{r})} + 2\frac{\nabla\mathcal{J}(\mathbf{r})}{\mathcal{J}(\mathbf{r})}\frac{\nabla\Sigma(\mathbf{r})}{\Sigma(\mathbf{r})} + \frac{\nabla^2\Sigma(\mathbf{r})}{\Sigma(\mathbf{r})} \tag{F.1}$$

450 The derivatives of the Jastrow factor can be obtained given its functional form, either by
451 providing analytical expression for the derivatives of the different kernels or by relying on au-
452 tomatic differentiation. Since the Jastrow kernels are $\mathbb{R} \to \mathbb{R}$ maps automatic evaluation of the
453 second derivative via double backpropagation does not lead to the calculation of unnecessary
454 mixed second derivative terms. The gradient of the determinant part can be expressed as :

$$\nabla\Sigma = \sum_n d_n \left(\frac{\nabla D_n^\uparrow}{D_n^\uparrow} + \frac{\nabla D_n^\downarrow}{D_n^\downarrow}\right) D_n^\uparrow D_n^\downarrow \tag{F.2}$$

455 where the gradient of the individual determinant terms can be obtained via the Jacobi
456 formula [62]:

$$\frac{\nabla D}{D} \equiv \frac{\nabla \det A}{\det A} = \text{Tr}(A^{-1}\nabla A) \tag{F.3}$$

457 The diagonal hessian of the determinant part $\Sigma$ is obtained via:

$$\nabla^2\Sigma = \sum_n d_n \left(\frac{\nabla^2 D_n^\uparrow}{D_n^\uparrow} + 2\frac{\nabla D_n^\uparrow}{D_n^\uparrow}\frac{\nabla D_n^\downarrow}{D_n^\downarrow} + \frac{\nabla^2 D_n^\downarrow}{D_n^\downarrow}\right) D_n^\uparrow D_n^\downarrow \tag{F.4}$$

458 Here the diagonal Hessian of the determinant terms can be obtained via the formula [62]:

$$\frac{\nabla^2 D}{D} \equiv \frac{\nabla^2 \det(A)}{\det(A)} = \text{Tr}(A^{-1}\nabla^2 A) + \left(\text{Tr}(A^{-1}\nabla A)\right)^2 - \text{Tr}\left(A^{-1}\nabla A A^{-1}\nabla A\right) \tag{F.5}$$

459 Note that in the case of single-electron orbital, i.e. in absence of backflow transformation,
460 the last two terms of eq. (F.5) cancel each other leading to : $\frac{\nabla^2 \det(A)}{\det(A)} = \text{Tr}(A^{-1}\nabla^2 A)$ as demon-
461 strated in [32] (see appendix G).
462

463 The use of the eq. (F.5) (or G.7 in absence of backflow transformation), leads to a signif-
464 icant speed up the of the calculation of the kinetic energy. As seen in Fig. 10, a speedup of
465 almost two orders of magnitude can be obtained for $CO$. Even larger speedups are expected
466 for larger molecules as the automatic differentiation approach scales poorly with the number
467 of electrons in the molecule.

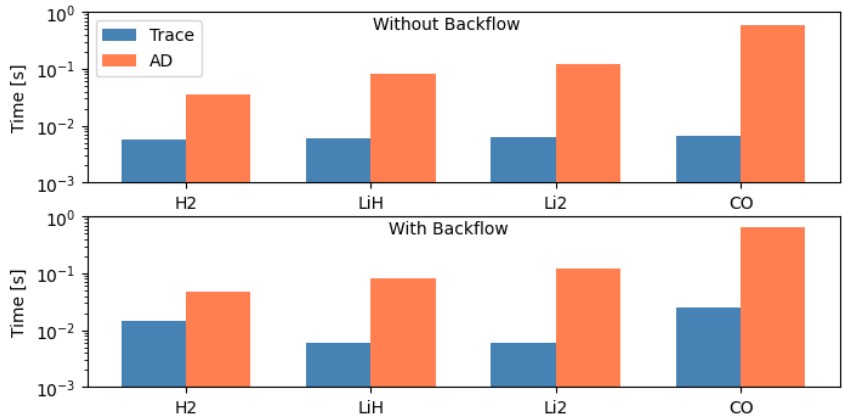

Figure 10: Runtime for the calculation of the kinetic energy using automatic differentiation (AD) or the matrix trace formulas (Trace). The runtime was computed on an NVIDIA A100 for four different molecules, using only the ground state in the wavefunction expansion, 1000 sampling points and with/without backflow transformation using eqs. (F.5) and (G.7) respectively

### F.2 Derivative of Excited-State Determinants

Calculating the first and second determinant derivatives given by eqs. (F.2) and (F.4) requires the inversion of a large number of $A$ matrices corresponding to the different determinants. As the number of determinants increases, this becomes a computational bottleneck. The determinants included in the wave function (4) correspond to well defined electronic configurations. The determinants contained in the wave function only differ from the ground state determinant by the exchange of a finite number of columns (1 for single-electron excitations, 2 for double-electron excitations etc ...). This relationship allows to avoid the calculation of many matrix inverses.

Let's consider a reference determinant $D_0 = \det A_0$, an excited determinant $D_X = \det A_X$, a 1 body operator $\mathcal{O}$ (e.g. $\nabla$ or $\nabla^2$) with $B = \mathcal{O}A$. The matrix of molecular orbitals can be written as $\mathbf{\Phi} = [A_0|A_v]$ where $A_v$ is the rectangular matrix of the virtual molecular orbitals. The matrix $A_X$ is consequently constructed by replacing some columns of $A_0$ by columns from $A_v$. One can then show that [32]:

$$\text{Tr}\left(A_X^{-1}B_X\right) = \text{Tr}(A_0^{-1}B_0) + \text{Tr}\left((PA_0^{-1}A_XP)^{-1} \cdot PMP\right) \tag{F.6}$$

with:

$$M = A_0^{-1}B_X - A_0^{-1}B_0A_0^{-1}A_X \tag{F.7}$$

Similarly one can show that (see Appendix H):

$$
\begin{aligned}
\text{Tr}\left(\left(A_X^{-1}B_X\right)^2\right) &= \text{Tr}\left((A_0^{-1}B_0)^2\right) \\
&+ \text{Tr}\left(\left((PA_0^{-1}A_XP)^{-1} \cdot PMP\right)^2\right) \\
&+ 2\text{Tr}\left((PA_0^{-1}A_XP)^{-1} \cdot PYP\right)
\end{aligned}
\tag{F.8}
$$

with :

$$Y = A_0^{-1}B_0M \tag{F.9}$$

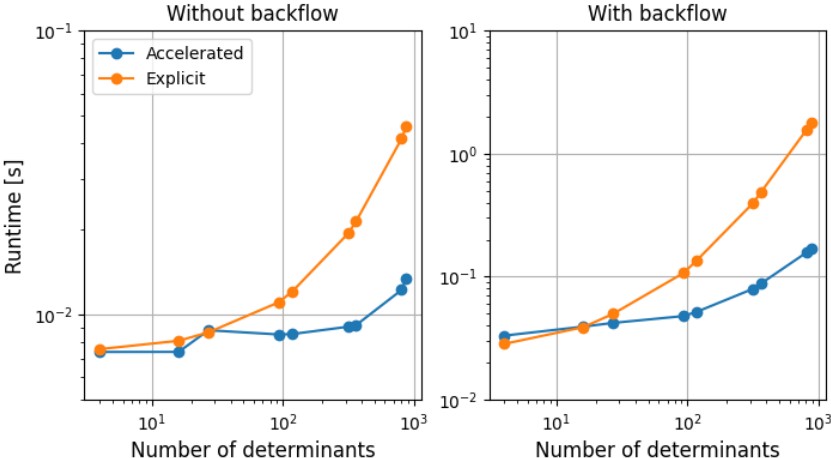

Figure 11: Runtime for the calculation of the kinetic energy of a *CO* molecule using 1000 walkers and different electronic configurations. The calculations were done on an NVIDIA A100 GPU and with and without backflow transformation. Two approaches are compared: Explicit - using eq (F.5) (or (G.7) without backflow) on all the determinants in the wave function; Accelerated - using eqs. (F.6) and (F.8) to compute the contribution of the excited state determinants of the wavefunction.

and $P$ the projector on the space of columns which are different in $A_0$ and $A_x$ [32]. Note that in eq. F.8, the convention: $(AB)^2 = ABAB$ was used to shorten the expressions. As seen on the equations (F.6) and (F.8), $A_0$ is the only large matrix whose inverse needs to be computed. The other inverses, namely $(PA_0^{-1}A_X P)^{-1}$, are in our case only of dimension 1 or 2 as we only consider single and double excitations. This greatly accelerate the calculations of the different terms present in eq. (F.2) and (F.4).

As shown in Fig. 11, using eqs. (F.6) and (F.8) to compute the kinetic energy leads to a one order of magnitude speedup compared to computing all the determinants explicitly. Note however that if only a small number of determinants are included in the wavefunction expansion, the explicit calculation might be faster or at least competitive with the accelerated method.

# G    Simplification of Eq. F.5 in Absence of Backflow Transformation

As explained in the main text, eq. F.5 reduces to the expression derived by Filippi et al. [32] in the case of single-electron orbitals, i.e. $A_{i,j} = \phi_{k_i}(r_j)$. In such a case the derivative of $A$ w.r.t a given electronic position is given by:

$$\frac{\partial A}{\partial x_j} = \begin{pmatrix} 0 & \dots & 0 \\ \vdots & & \vdots \\ \frac{\partial \phi_{k_1}(r_j)}{\partial x_j} & \dots & \frac{\partial \phi_{k_n}(r_j)}{\partial x_j} \\ \vdots & & \vdots \\ 0 & \dots & 0 \end{pmatrix} \tag{G.1}$$

We therefore have:

$$\text{Tr}\left(A^{-1}\frac{\partial A}{\partial x_j}\right) = \text{Tr}(uv^T) = v^T \cdot u \equiv c \tag{G.2}$$

where $u$ and $v$ are two column vectors representing respectively the j-th column of $A^{-1}$ and $v$ the j-th row of $\frac{\partial A}{\partial x_j}$. We therefore have:

$$\left(\text{Tr}\left(A^{-1}\frac{\partial A}{\partial x_j}\right)\right)^2 = c^2 \tag{G.3}$$

Similarly we can show that:

$$\text{Tr}\left(A^{-1}\frac{\partial A}{\partial x_j}A^{-1}\frac{\partial A}{\partial x_j}\right) = \text{Tr}(u(v^Tu)v^T) = c \cdot \text{Tr}(uv^T) = c^2 \tag{G.4}$$

Combining the last two equations we have:

$$\left(\text{Tr}\left(A^{-1}\frac{\partial A}{\partial x_j}\right)\right)^2 = \text{Tr}\left(A^{-1}\frac{\partial A}{\partial x_j}A^{-1}\frac{\partial A}{\partial x_j}\right) \tag{G.5}$$

We can easily extend that results to show that:

$$\left(\text{Tr}(A^{-1}\nabla A)\right)^2 = \text{Tr}(A^{-1}\nabla A A^{-1}\nabla A) \tag{G.6}$$

As a consequence, for single-electron molecular orbital the two last terms of eq. F.5 cancel each other and one obtain the result of Filippi et al [32]:

$$\frac{\nabla^2 D}{D} = \text{Tr}(A^{-1}\nabla^2 A) \tag{G.7}$$

# H Derivation of Eqs. F.6 and F.8

Equation F.6 has already been demonstrated by Filippi et al. [32]. We demonstrate it again following a slightly different method to facilitate the demonstration of eq. F.8.

Let's define $A_0$ and $A_X$ as the molecular orbital matrix of the ground state determinant and of a given excited state determinant respectively. The two matrices only differ by a few columns of $A_0$ that have been replaced by column vectors of virtual orbitals. We also define the projector $P$ on the subspace of these exchanged columns. $P$ is a diagonal matrix with ones at the indices of the exchanged columns and zeros elsewhere. We also define the complementary projector $Q$ such as $P + Q = \mathbb{I}$. Finally we define the matrices $B_0$ and $B_X$ as the application of a 1-body operator (either $\nabla$ or $\nabla^2$) on $A_0$ and $A_X$.

As shown in [32] $A_X(B_X)$ can be expressed as :

$$A_X = A_0 + (A_X - A_0)P \tag{H.1}$$

and

$$B_X = B_0 + (B_X - B_0)P \tag{H.2}$$

### H.1 Inverse of $A_X$

Equations F.6 and F.8 involve the inverse of the excited state matrix $A_X$. Using eq. H.1 we can write this inverse as:

$$A_X^{-1} = (A_0 + (A_X - A_0)P)^{-1} \tag{H.3}$$

We can use the Kailath variant of the Woodbury matrix inverse formulas [62]:

$$(U + VW)^{-1} = U^{-1} - U^{-1}V(I + WU^{-1}V)^{-1}WU^{-1} \tag{H.4}$$

with $U = A_0$, $V = (A_X - A_0)$ and $W = P$ to express the inverse of $A_X$ :

$$
\begin{aligned}
A_X^{-1} &= A_0^{-1} - A_0^{-1}(A_X - A_0)\left(\mathbb{I} + PA_0^{-1}(A_X - A_0)\right)^{-1} PA_0^{-1} & \text{(H.5)} \\
&= A_0^{-1} - \left(A_0^{-1}A_X - \mathbb{I}\right)\left(\mathbb{I} + PA_0^{-1}A_X - P\right)^{-1} PA_0^{-1} & \text{(H.6)} \\
&= A_0^{-1} - \left(A_0^{-1}A_X - \mathbb{I}\right)\left(Q + PA_0^{-1}A_X\right)^{-1} PA_0^{-1} & \text{(H.7)} \\
&= A_0^{-1} - \left(A_0^{-1}A_X - \mathbb{I}\right)\left(Q + PA_0^{-1}A_X P\right)^{-1} PA_0^{-1} & \text{(H.8)}
\end{aligned}
$$

The equation H.8 is obtained by noting that following H.1 we have: $A_0^{-1}A_X = A_0^{-1}A_X P$ and therefore: $PA_0^{-1}A_X = PA_0^{-1}A_X P$.

The term $\left(Q + PA_0^{-1}A_X P\right)$ in eq. H.8 is a block matrix on the subspaces spanned by the $P$ and $Q$ and its inverse can therefore be written as [62]

$$\left(Q + PA_0^{-1}A_X P\right)^{-1} = Q + \left(PA_0^{-1}A_X P\right)^{-1} = Q + P\left(A_0^{-1}A_X\right)^{-1} P \tag{H.9}$$

Note that the last equality is only valid since $QA_0^{-1}A_X Q = Q$. Since $QP = 0$ and $P^2 = P$ by definition, eq. H.8 can then be simplified to:

$$A_X^{-1} = A_0^{-1} - \left(A_0^{-1}A_X - \mathbb{I}\right)\left(P(A_0^{-1}A_X)^{-1}P\right)A_0^{-1} \tag{H.10}$$

In the following we use the following definitions:

$$T = \left(P(A_0^{-1}A_X)^{-1}P\right) \tag{H.11}$$

and

$$Z = \left(A_0^{-1}A_X - \mathbb{I}\right)TA_0^{-1} \tag{H.12}$$

### H.2 Demonstration of eq. F.6

We are trying here to find a convenient expression for:

$$\mathrm{Tr}\left(A_X^{-1}B_X\right) \tag{H.13}$$

that does not require to compute the inverse of all the possible $A_X$ matrices contained in the CI expansion considered in the definition of the wave function. Using eq. H.10, equation H.13 can therefore be written as :

$$
\begin{aligned}
\mathrm{Tr}\left(A_X^{-1}B_X\right) &= \mathrm{Tr}\left(A_0^{-1}B_X\right) - \mathrm{Tr}\left(\left(A_0^{-1}A_X - \mathbb{I}\right)TA_0^{-1}B_X\right) \\
&= \mathrm{Tr}\left(A_0^{-1}B_X\right) - \mathrm{Tr}\left(A_0^{-1}A_X TA_0^{-1}B_X\right) + \mathrm{Tr}\left(TA_0^{-1}B_X\right)
\end{aligned}
$$

543    Using eq. H.2 and the cyclic property of the trace, we can further obtain:

$$
\begin{aligned}
\mathrm{Tr}\left(A_X^{-1}B_X\right) &= \mathrm{Tr}\left(A_0^{-1}B_0\right) + \mathrm{Tr}\left(A_0^{-1}(B_X - B_0)P\right) \\
&\quad - \mathrm{Tr}\left(TA_0^{-1}B_0 A_0^{-1}A_X\right) \\
&\quad - \mathrm{Tr}\left(TA_0^{-1}(B_X - B_0)PA_0^{-1}A_X\right) \\
&\quad + \mathrm{Tr}\left(TA_0^{-1}B_X\right)
\end{aligned}
$$

544    The fourth term of this equation can be simplified since $PA_0^{-1}A_X = PA_0^{-1}A_X P$ and as demon-
545    strated in [32]

$$
\left(P(A_0^{-1}A_X)P\right)\left(P(A_0^{-1}A_X)^{-1}P\right) = P \tag{H.14}
$$

546    Consequently the forth term can be simplified to

$$
\begin{aligned}
\mathrm{Tr}\left(\left(P(A_0^{-1}A_X)^{-1}P\right)A_0^{-1}(B_X - B_0)PA_0^{-1}A_X\right) &= \mathrm{Tr}\left(\left(P(A_0^{-1}A_X)^{-1}P\right)A_0^{-1}(B_X - B_0)PA_0^{-1}A_X P\right) \\
&= \mathrm{Tr}\left(A_0^{-1}(B_X - B_0)PA_0^{-1}A_X P\left(P(A_0^{-1}A_X)^{-1}P\right)\right) \\
&= \mathrm{Tr}\left(A_0^{-1}(B_X - B_0)P\right)
\end{aligned}
$$

547    and can be simplified with the second term of the equation. Factorizing the terms we arrive
548    at the following equation:

$$
\begin{aligned}
\mathrm{Tr}\left(A_X^{-1}B_X\right) &= \mathrm{Tr}\left(A_0^{-1}B_0\right) \\
&\quad + \mathrm{Tr}\left(\left(P(A_0^{-1}A_X)^{-1}P\right)\left(A_0^{-1}B_X - A_0^{-1}B_0 A_0^{-1}A_X\right)\right)
\end{aligned}
$$

### H.3   Demonstration of eq. F.8

550    We are trying here to derive a convenient expression for:

$$
\mathrm{Tr}\left(A_X^{-1}B_X A_X^{-1}B_X\right) \equiv \mathrm{Tr}\left((A_X^{-1}B_X)^2\right) \tag{H.15}
$$

551    Using eq. H.10, i.e. $A_X^{-1} = A_0^{-1} - Z$, we can unroll the square to obtain:

$$
\mathrm{Tr}\left((A_X^{-1}B_X)^2\right) = \underbrace{\mathrm{Tr}\left((A_0^{-1}B_X)^2\right)}_{\alpha} + \underbrace{\mathrm{Tr}\left((ZB_X)^2\right)}_{\gamma} - 2\underbrace{\mathrm{Tr}\left(A_0^{-1}B_X ZB_X\right)}_{\beta} \tag{H.16}
$$

552    We start by injecting eq. H.2 in the fist term of the equation above to obtain:

$$
\mathrm{Tr}\left((A_0^{-1}B_X)^2\right) = \underbrace{\mathrm{Tr}\left((A_0^{-1}B_0)^2\right)}_{\alpha_1} + \underbrace{\mathrm{Tr}\left((A_0^{-1}B_\Delta)^2\right)}_{\alpha_2} + 2\underbrace{\mathrm{Tr}\left(A_0^{-1}B_\Delta A_0^{-1}B_0\right)}_{\alpha_3} \tag{H.17}
$$

553    with $B_\Delta = (B_X - B_0)P = B_X - B_0$. Using the definition of $Z$, the second term of eq. H.16
554    leads to:

$$
\mathrm{Tr}\left((ZB_X)^2\right) = \underbrace{\mathrm{Tr}\left((A_0^{-1}A_X TA_0^{-1}B_X)^2\right)}_{\gamma_1} + \underbrace{\mathrm{Tr}\left((TA_0^{-1}B_X)^2\right)}_{\gamma_2} - 2\underbrace{\mathrm{Tr}\left((TA_0^{-1}B_X)^2 A_0^{-1}A_X\right)}_{\gamma_3} \tag{H.18}
$$

Using the definition of $B_X$, the cyclic property of the trace and the identity: $PA_0^{-1}A_XT = P$ this can be simplified to:

$$\gamma_1 = \underbrace{\mathrm{Tr}\left((A_0^{-1}A_XTA_0^{-1}B_0)^2\right)}_{\gamma_{11}} + \underbrace{\mathrm{Tr}\left((A_0^{-1}B_\Delta)^2\right)}_{\gamma_{12}} + 2\underbrace{\mathrm{Tr}\left(A_0^{-1}B_0A_0^{-1}A_XTA_0^{-1}B_\Delta\right)}_{\gamma_{13}} \tag{H.19}$$

$$\gamma_2 = \underbrace{\mathrm{Tr}\left((TA_0^{-1}B_0)^2\right)}_{\gamma_{21}} + \underbrace{\mathrm{Tr}\left((TA_0^{-1}B_\Delta)^2\right)}_{\gamma_{22}} + 2\underbrace{\mathrm{Tr}\left(TA_0^{-1}B_0TA_0^{-1}B_\Delta\right)}_{\gamma_{23}} \tag{H.20}$$

$$\gamma_3 = \overbrace{\mathrm{Tr}\left((TA_0^{-1}B_0)^2A_0^{-1}A_X\right)}^{\gamma_{31}} + \overbrace{\mathrm{Tr}\left((A_0^{-1}B_\Delta)^2T\right)}^{\gamma_{32}}$$
$$+ \underbrace{\mathrm{Tr}\left(A_0^{-1}B_0TA_0^{-1}B_\Delta\right)}_{\gamma_{33}} + \underbrace{\mathrm{Tr}\left(TA_0^{-1}B_\Delta TA_0^{-1}B_0A_0^{-1}A_X\right)}_{\gamma_{34}}$$

Finally the last term can be expanded to:

$$\mathrm{Tr}\left(A_0^{-1}B_XZB_X\right) = \underbrace{\mathrm{Tr}\left(A_0^{-1}B_XA_0^{-1}A_XTA_0^{-1}B_X\right)}_{\beta_1} - \underbrace{\mathrm{Tr}\left(A_0^{-1}B_XTA_0^{-1}B_X\right)}_{\beta_2} \tag{H.21}$$

which can be expanded to:

$$\beta_1 = \overbrace{\mathrm{Tr}\left((A_0^{-1}B_0)^2A_0^{-1}A_XT\right)}^{\beta_{11}} + \overbrace{\mathrm{Tr}\left((A_0^{-1}B_\Delta)^2\right)}^{\beta_{12}}$$
$$+ \underbrace{\mathrm{Tr}\left(A_0^{-1}B_0A_0^{-1}A_XTA_0^{-1}B_\Delta\right)}_{\beta_{13}} + \underbrace{\mathrm{Tr}\left(A_0^{-1}B_\Delta A_0^{-1}B_0\right)}_{\beta_{14}}$$

and

$$\beta_2 = \underbrace{\mathrm{Tr}\left((A_0^{-1}B_0)^2T\right)}_{\beta_{21}} + \underbrace{\mathrm{Tr}\left((A_0^{-1}B_\Delta)^2T\right)}_{\beta_{22}} + \underbrace{\mathrm{Tr}\left(A_0^{-1}B_\Delta A_0^{-1}B_0T\right)}_{\beta_{23}} + \underbrace{\mathrm{Tr}\left(A_0^{-1}B_0A_0^{-1}B_\Delta T\right)}_{\beta_{24}} \tag{H.22}$$

Some of the above terms cancel each other:

$$\alpha_2 + \gamma_{12} - 2\beta_{12} = 0 \tag{H.23}$$
$$\alpha_3 - \beta_{13} = 0 \tag{H.24}$$
$$\gamma_{32} - \beta_{22} = 0 \tag{H.25}$$
$$\gamma_{33} - \beta_{23} = 0 \tag{H.26}$$

and some of the terms can be re-written:

$$\gamma_{11} + \gamma_2 - 2(\gamma_{31} + \gamma_{34}) = \mathrm{Tr}\left((TM)^2\right) \tag{H.27}$$
$$\beta_{21} + \beta_{24} = \mathrm{Tr}\left(A_0^{-1}B_0A_0^{-1}B_XT\right) \tag{H.28}$$

which after some re-factorization leads to:

$$\mathrm{Tr}\left(A_X^{-1}B_XA_X^{-1}B_X\right) = \mathrm{Tr}\left((A_0^{-1}B_0)^2\right) + \mathrm{Tr}\left((MT)^2\right) + 2\mathrm{Tr}\left(A_0^{-1}B_0MT\right)$$

Note that here the $^2$ means $(AB)^2 = ABAB$

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
