# Peer review of "QMCTorch: Molecular Wavefunctions with Neural Components for Energy and Force Calculations"

_SciPost Chemistry_

## Round 1 · Referee Report · Anonymous (Referee 1) · 2025-7-10

Strengths

• The manuscript gives an in-depth overview of Quantum Monte Carlo in combination with neural networks, within the QMCTorch program. It uses external programs (ADF, PySCF) for providing baseline data.
• The results from QMCTorch are recovering a large portion of electron correlation.

Weaknesses

• The figures are not ordered correctly. I.e. Figures 6 and 7 are referenced before Figs 3-5. Also, they should be placed close to the point in the main text where they are referenced.
• Table 1 gives results with Paulinet only for H2 and LiH, but not for Li2 and CO; vice versa, Ferminet results are only given for Li2 and CO, but not for H2 and LiH. No explanation nor discussion is given for this. This needs to be added.
• The portion of electron correlation that is recovered is system-dependent. Is this an indication of multi-reference character of the wavefunction?
• Annex A reports a comparison between GPUs and CPUs, but it is not clear if the same accuracy is obtained. I.e. in standard QM programs one can tune the accuracy of the integrals, and evaluate the accuracy of the densities and energies, to a certain cutoff before convergence is achieved. It would be good to add a discussion in the Annex A about the accuracy.

Report

Overall it is an interesting new program with good results. Some discussion is needed on the portion of electron correlation, and whether a more or less constant portion could be achievable.

Requested changes

1- Figures need to be reordered, with numbers reflecting the ordering as they are referenced within the main text. Figures need to be positioned close to where they are referenced. 2- Discussion of absence of Paulinet and Ferminet results for some molecules need to be added 3- Discussion of the portion of electron correlation obtainable, and whether this is depending on multi-reference character, needs to be added. 4- Discussion of accuracy obtained with CPUs vs. GPUs needs to be added.

Recommendation

Ask for minor revision

---

## Round 1 · Referee Report · Anonymous (Referee 2) · 2025-7-12

Report

The paper provides an extended introduction to the QMCtorch package, which was first introduced in reference 29 by the same author. QMCtorch performs quantum Monte Carlo calculations using neural network wavefunctions. Neural network-based quantum Monte Carlo has emerged as a highly promising and accurate method for calculating the many-body wavefunctions of molecules based on unsupervised learning. In recent years, several research groups have developed packages on different platforms. As the method is still evolving rapidly, the development of QMCtorch is a valuable addition to the community, offering a new platform for developers based on the Torch framework. Therefore, I recommend that this paper is suitable for publication in SciPost Chemistry, satisfying one of the acceptance criteria “Open a new pathway in an existing or a new research direction, with clear potential for multi-pronged follow-up work”.

Additionally, I would like to offer a few suggestions for the authors to improve the manuscript, thereby benefiting readers interested in entering this field.
1. On Page 3: “Two notable machine learning-inspired approaches, Ferminet [24] and Paulinet [25], have demonstrated remarkable accuracy in computing ground-state energies of molecular systems”. While Ferminet and Paulinet are indeed seminal works in the field, they are no longer the most powerful networks available today. Latest developments such as Psiformer and LapNet (and several new arxiv preprints of this year) should be mentioned to help readers stay abreast of recent progress.
2. The current experiments are limited to very small systems. While this is understandable for a paper introducing a new code, it would be beneficial if the authors could also mention that neural network calculations can now be applied to molecules with over 100 electrons.
3. The manuscript could be enhanced if the authors discuss the potential for implementing periodic boundary conditions, as has been done by several other groups.
4. The current experiments use only 500 optimization steps, and the plots indicate that convergence has not yet been achieved. More training steps are needed to ensure that the reported results are converged.
5. QMCtorch yields suboptimal results for some of the tested systems (e.g., only 63% for CO), which is significantly lower than those obtained by other state-of-the-art works. An explanation or commentary on this discrepancy is necessary.
6. The paper notes that the optimization of neural-network Jastrow factors and backflow transformations is noisy and provides limited performance improvements. It would be valuable to explore the underlying reasons for this (e.g., network architecture, initialization, or training strategy).
7. Interatomic force calculations using neural networks have been reported in previous studies, where several different estimators are tested and new estimators and sampling methods are proposed for specific neural networks. The author may wish to compare their results with these studies and test different force estimators to further improve the accuracy and efficiency of force calculations.

Recommendation

Ask for minor revision

---

## Round 1 · Referee Report · Anonymous (Referee 3) · 2025-7-15

Report

The present paper reports an overview of QMCTorch, a QMC framework based on the deep learning library PyTorch. I have read Reviewer #2’s report and found his/her comments to be highly relevant. In the following, I offer a complementary set of observations that I hope will help improve the quality and clarity of the manuscript. I would like to preface my remarks by noting that I have limited expertise in ML, AI, and related techniques. My comments, therefore, primarily concern the QMC methodology employed in the work and the corresponding numerical results.

I am not convinced that the present work presents a significant level of novelty compared to Ref. [29]. Moreover, the numerical results are not particularly compelling. I also have several concerns regarding both the numerical findings and the manuscript as a whole. Accordingly, I am not recommending publication of the present manuscript in SciPost Physics.

  • The beginning of the introduction (up to line 52) provides a very general overview of QMC. However, the specific approach used here is variational Monte Carlo (VMC), which is only one flavor of QMC. I suggest the authors refer to their method explicitly as VMC rather than QMC to avoid confusion and to improve precision.

  • In line 57, the authors cite Filippi and co-workers regarding QMC studies of excited states. However, there is a substantial body of prior work by Caffarel, Scemama, Loos, and collaborators in this area. See, for example:

  • J. Chem. Theory Comput. 14, 1395 (2018),
  • J. Chem. Phys. 149, 034108 (2018),
  • Res. Chem. 1, 100002 (2019),
  • J. Chem. Phys. 153, 174107 (2020). I believe these important contributions should be acknowledged. Currently, the manuscript seems somewhat overly focused on the work of Filippi.

  • The manuscript states that "QMCTorch includes a fully differentiable atomic-orbital layer that computes the AO values at the given electronic coordinates." Out of curiosity, does this implementation rely on QMCkl (https://github.com/TREX-CoE/qmckl)? If not, integrating QMCkl could significantly improve computational efficiency.

  • While automatic differentiation offers great flexibility, it generally comes at a performance cost compared to an analytic formulation. Can the author provide a rough estimate of the computational overhead introduced by using automatic differentiation?

  • The author notes that "In contrast to other approaches [25], QMCTorch does not require a preliminary calculation to determine the most relevant determinants to include in the expansion." This is an interesting feature, but it truly deserves further elaboration. What is the selection procedure? At the moment, it looks like magic!

  • I find the results in Figure 2 puzzling. As the trial wave function improves (from left to right), the fluctuations in the correlation energy increase. This is counterintuitive and appears systematic across all systems studied. How should one interpret this trend? Am I missing an important aspect of the methodology or implementation?

  • Table 1 shows that QMCTorch consistently recovers a smaller fraction of the correlation energy compared to PauliNet and FermiNet. The author attributes this to a limited number of optimization steps, fewer variational parameters, and a less effective optimizer. However, these points merit deeper investigation. I believe the author should perform additional tests to evaluate the effect of these factors more systematically, either by further QMCTorch runs or comparisons with PauliNet/FermiNet under controlled settings. As it stands, this is a weak point of the study.

  • I struggled to understand the purpose and relevance of Section 3.2 within the overall structure of the manuscript. It is not clear to me how this section contributes to the main objectives of the paper. Clarification would be welcome.

  • The comparison of energies and forces between QMCTorch and CCSD in Section 3.3 raises concerns. The CCSD results are obtained with a DZP Gaussian basis, and such a comparison may not be meaningful given the fundamentally different nature of the Hilbert spaces involved. In this context, is CCSD/DZP a suitable reference?

Recommendation

Reject

---

## Editorial Decision

awaiting_resubmission